# Effects of PM_2.5_ on Cardio-Pulmonary Function Injury in Open Manganese Mine Workers

**DOI:** 10.3390/ijerph16112017

**Published:** 2019-06-06

**Authors:** Yuanni Huang, Mian Bao, Jiefeng Xiao, Zhaolong Qiu, Kusheng Wu

**Affiliations:** Department of Preventive Medicine, Shantou University Medical College, Shantou 515041, China; yuannihuang@163.com (Y.H.); 16mbao1@stu.edu.cn (M.B.); 18934167489@163.com (J.X.); 18zlqiu@stu.edu.cn (Z.Q.)

**Keywords:** fine particulate matter 2.5 (PM_2.5_), pulmonary function, cardiac function, blood pressure, electrocardiogram abnormalities, open-pit manganese mine

## Abstract

Exposure to fine particulate matter 2.5 (PM_2.5_) is associated with adverse health effects, varying by its components. The health-related effects of PM_2.5_ exposure from ore mining may be different from those of environment pollution. The aim of this study was to investigate the effects of different concentrations of PM_2.5_ exposure on the cardio-pulmonary function of manganese mining workers. A total of 280 dust-exposed workers who were involved in different types of work in an open-pit manganese mine were randomly selected. According to the different concentrations of PM_2.5_ in the working environment, the workers were divided into an exposed group and a control group. The electrocardiogram, blood pressure, and multiple lung function parameters of the two groups were measured and analyzed. The PM_2.5_ exposed group had significantly lower values in the pulmonary function indexes of forced expiratory volume in one second (FEV1.0), maximum mid expiratory flow (MMEF), peak expiratory flow rate (PEFR), percentage of peak expiratory flow out of the overall expiratory flow volume (PEFR%), forced expiratory flow at 25% and 75% of forced vital capacity (FEF 25, FEF75), forced expiratory flow when 25%, 50%, and 75% of forced vital capacity has been exhaled (FEF25%,FEF50%, FEF75%), and FEV1.0/FVC% (the percentage of the predicted value of forced vital capacity) than the control group (all *p* < 0.05). Both groups had mild or moderate lung injury, most of which was restrictive ventilatory disorder, and there was significant difference in the prevalence rate of restrictive respiratory dysfunction between the two groups (41.4% vs. 23.6%, *p* = 0.016). Electrocardiogram (ECG) abnormalities, especially sinus bradycardia, were shown in both groups, but there was no statistical difference of the prevalence rate between the two groups (*p* > 0.05). Also, no significant difference of the prevalence rate of hypertension was observed between the PM2.5 exposure and control groups (*p* > 0.05). PM_2.5_ exposure was associated with pulmonary function damage of the workers in the open-pit manganese mine, and the major injury was restrictive ventilatory disorder. The early effect of PM_2.5_ exposure on the cardiovascular system was uncertain at current exposure levels and exposure time.

## 1. Introduction

Mined ore needs to be broken, screened, and beneficiated to obtain the necessary minerals. The production process is often accompanied by direct or indirect emissions of particulars of different diameters. All kinds of fine particulate matter pollution occur in the process of mining, crushing, ball milling, and so on, through which it is easy to form a variety of aerosol particle sizes. The formation of fine particulate matter pollution concentrates heavy metals, organic pollutants, etc., and can be directly entered into the alveoli and the blood circulatory system. Fine particulate matter pollution is a type of air pollution that comprises a heterogeneous mixture of different particle sizes and chemical compositions. Fine particulate matter aerosol (PM_2.5_ with the diameter ≤ 2.5 μm) is characterized by its small particle size and its large surface area and toxin absorption ability [1]. These properties make it possible for PM_2.5_ to invade the smallest airways, including alveolar tissue.

Airborne fine particulate matter (PM_2.5_) is continuing to attract more and more attention due to its environmental effects. There are various sources of PM_2.5_, and the components may also have different effects on people. PM_2.5_ comprises a mixture of solid and liquid particles, including black carbon, metals, nitrate, sulfate, polycyclic aromatic hydrocarbons, and automobile exhaust particles [2]. The chemical properties and pathological toxicity of PM_2.5_ are known to influence a variety of respiratory diseases [3]. Suggested mechanisms of PM_2.5_ that lead to adverse effects and chronic diseases include increasing oxidative stress, inflammatory responses, and genotoxicity [3,4]. Inhaled PM_2.5_ can deposit in different compartments in the respiratory tract and interact with epithelial cells and resident immune cells. Exposure to PM_2.5_ can induce local or systematic inflammatory responses [5].

The potential health effects of PM_2.5_ currently represent some of the most important air quality issues. A large number of epidemiological studies have shown that chronic exposure to PM_2.5_ can cause damage to cardiovascular, respiratory, and endocrine systems [6,7]. Other studies have also showed that PM_2.5_ can induce a variety of chronic diseases, such as respiratory system damage, cardiovascular dysfunction, and diabetes mellitus [3]. Previous studies have identified a marked association between PM_2.5_ exposure and increased incidence of pulmonary diseases [8]. One study conducted in China suggested that PM_2.5_ in 2015 contributed as much as 40.3% to total stroke deaths, 33.1% to acute lower respiratory infection (<5yr) deaths, 26.8% to ischemic heart disease (IHD) deaths, 23.9% to lung cancer (LC) deaths, 18.7% to chronic obstructive pulmonary disease (COPD) deaths, 30.2% to total deaths combining IHD, stroke, COPD, and LC, and 15.5% to all-cause deaths [9]. The number of peripheral arterial disease patients was reported to increase by 0.26% for each 10 μg/m^3^ increase in acute PM_2.5_ exposure in the northeastern United States [10]. Similarly, in Beijing, China, PM_2.5_ is significantly correlated with the number of emergency room visits for cardiovascular diseases (especially IHD, arrhythmia, and atrial fibrillation) [11]. Another study showed high prevalence of respiratory complaints in Congolese informal coltan miners, and an inverse association was observed between lung function (peak expiratory flow rate, PEFR) and PM_2.5_ exposure [12].

PM_2.5_ is known to cause epigenetic and microenvironmental alterations in lung cancer, including tumor-associated signaling pathway activation mediated by microRNA dysregulation, DNA methylation, and increased levels of cytokines and inflammatory cells [3]. Previous research has identified that air pollution is associated with various respiratory diseases, but few studies have investigated the respiratory function served by PM_2.5_ in these diseases [3]. The effects of PM_2.5_ exposure on the cardio-pulmonary function of mine workers have also been scarcely investigated.

While numerous studies have demonstrated that exposure to fine particulate matter pollution is associated with adverse health effects, the characteristics of fine particulate matter pollution that cause harm are not well understood, and fine particulate matter pollution toxicity may vary by its chemical composition [13]. The properties of PM_2.5_ often vary with environmental changes, and therefore further clarification is required [3]. The chemical compositions of PM_2.5_ from air pollution and mining may be also different. The production of fine particulate matter pollution from mines is mainly derived from the ore mining, crushing, ball grinding processes, etc. Under the influence of fine particulate matter pollution, the mine staff staying there for long-term periods are facing health risks in the working environment. Investigating the impact of PM_2.5_ on respiratory and cardiovascular systems is essential, since numerous populations, including ours, are currently experiencing heavy pollution of PM_2.5_. This study chose a specific open-pit manganese ore plant as the research site, and recruited the PM-exposed workers as the target population in order to explore different concentrations of PM_2.5_ exposure from ore mining and its effects on the cardiopulmonary function of manganese ore workers.

## 2. Materials and Methods

### 2.1. Subjects

All the participants were recruited from an open manganese mine. The manganese mine is located in Raoping county, Chaozhou city, Guangdong province of China, with a latitude of 23.70° N and longitude of 116.93° E. The manganese mine is considered moderate-sized in China, with a total area of 150 acres, including several workshops, and during the duration of this study a total of 358 workers were engaged in different kinds of work in these workshops. The PM_2.5_ concentration of 8-h TWA (time-weighted average concentration) in these different workshops had been determined by TSI SIDEPAK AM510 Aerosol Monitor (TSI Inc., Shoreview, MN, USA), for 7 successive days according to the detailed instructions of the instrument. The 8-h TWA concentration of PM_2.5_ was calculated by the following formula:TWA = E/T_total_, E = C_1_T_1_ + C_2_T_2_ + …… + C_n_T_n_(1)
where T_total_ means the working time in a working day, and is 8 h in this study; E is the PM_2.5_ exposure dose under working hours; C_n_ is the corresponding concentration of contact in the time period of T_n_; and T_n_ is the corresponding contact duration under the concentration of C_n_. The PM_2.5_ concentration was determined for 7 successive days, therefore *n* is 7 in this study.

For sampling the participants, a two-stage cluster sampling method was employed in this study. In the first stage, the PM_2.5_ concentration of 8-h TWA in each workshop was measured. The workers were divided into an exposed group or a control group according to the PM_2.5_ concentration of the workshop where they served. Finally, the belt-driving workers, mineral-crushing workers, and mineral-sifting workers in the manganese mine with high PM_2.5_ exposure levels were recruited as the exposed group, while the mineral separation workers with low PM_2.5_ exposure were recruited as the control group. In the second stage, the workers in each workshop were numbered and sampled by random digital table method, and about 80 percent of the workers in each workshop were recruited to take part in the study. Inclusion criteria of the participants in this study were as follows: (1) no previous working history of dust exposure; (2) working in the open manganese mine for over one year. Exclusion criteria were as follows: individuals with a history of congenital heart disease, pulmonary fibrosis, pneumoconiosis, or other heart and lung diseases.

The general characteristics and health information were obtained from the health declaration of the workers’ on-the-job occupational health physical examination. Workers’ general information included name, gender, age, weight, height, body mass, smoking and dust work history, working environment, known respiratory diseases, and medical history, etc. All participants gave their informed written consent after receiving detailed explanations of the study and potential consequences prior to enrollment. This study was performed with the approval of the Human Ethical Committee of Shantou University Medical College (Approval no.: SUMC 2017XM-0051).

### 2.2. Occupational Health Examination

Occupational health examinations of the exposed and control workers were carried out by professional medical staff/doctors according to the Occupational Health Monitoring Technical Specification of China, and the examination items included lung function, electrocardiogram, and blood pressure. The health examination for all the participants was performed in the morning before working.

Blood pressure was measured with desktop mercury sphygmomanometer in the case of not using antihypertensive drugs and fasting. The measurements for right upper limb blood pressure were carried out three times in quiet environment with the participants sitting on a seat. The evaluating value was according to the diagnostic criteria of hypertension of the World Health Organization (WHO). Specifically, (120–139)/(80–89) mmHg (1 mmHg = 0.133 kPa) is regarded as the normal value. If the systolic pressure is greater than 140 mmHg and/or diastolic pressure is greater than 90 mmHg, it can be diagnosed as hypertension.

The routine 12-leads electrocardiogram (ECG) was performed by professional medical staff using an electrocardiograph (PC-80A, Heal Force, Shenzhen, China). Diagnostic criteria were defined as follows: (1) normal adult sinus rhythm frequency >100 times/min was regarded as sinus tachycardia; (2) the frequency of sinus rhythm <60 times/min was regarded as sinus bradycardia; (3) cases where the origin of sinus rhythm was unchanged, but the rhythm was irregular, and the P-P interphase difference in the same leads was >0.12s were regarded as sinus arrhythmia.

Spirometric measurements were conducted in the morning before working by professional medical workers using a portable pulmonary function test apparatus (AS-507, Minato Inc., Tokyo, Japan). For the evaluation of the lung function change of the workers, the following 13 items were used as lung function test indicators in this study. (1) Forced vital capacity (FVC) is the maximum amount of air that can be exhaled as soon as possible after maximum inhalation. (2) FVC% represents the percentage of the predicted value of forced vital capacity, which means the ability to exhale air of the measured lung capacity at the most rapid speed. (3) Forced expiratory volume in one second (FEV1.0) means the amount of air that is exhaled at the fastest rate after a deep inhalation. (4) FEV1.0% represents the percentage of the forced expiratory volume in 1 second out of the predicted value of the measured lung capacity. (5) FEV1.0/FVC% represents the percentage of forced expiratory volume of the first 1 second out of the forced vital capacity. The criteria for the classification of pulmonary ventilation dysfunction can be defined as follows: FVC < 80 and FEV1.0 > 70, restricted ventilation; FVC > 80 and FEV1.0 < 70, obstructive ventilation; FVC < 80 and FEV1.0 < 70, mixed ventilatory disorder. Pulmonary ventilation function damage was diagnosed according to the labor ability appraisal ‘Worker’s occupational injury and disability grade of occupational disease’ (GB/T16180-2014, China). (6) Peak expiratory flow rate (PEFR) refers to the fastest instantaneous velocity of expiratory flow in the forced vital capacity determination process. (7) PEFR% means the percentage of peak expiratory flow out of the overall expiratory flow volume. (8) Maximum mid expiratory flow (MMEF) is the average flow rate of 25% to 75% of the forced expiratory capacity calculated by the FVC curve. (9) Maximum ventilatory volume (MVV) is the maximum amount of air that can be breathed in a unit of time. The following other small airway analysis indexes were included: (10) FEF25 refers to forced expiratory flow after 25 percent of the vital capacity has been expelled. (11) FEF50 refers to forced expiatory flow after 50 percent of the vital capacity has been expelled. (12) FEF75 refers to forced expiratory flow after 75 percent of the vital capacity has been expelled. (13) FEF50/FEF25 refers to the ratio of FEF50 to FEF25.

### 2.3. Statistical Analyses

Statistical analyses were conducted with Microsoft Excel 2007 (Microsoft Inc., Redmond, WA, USA) and SPSS 23.0 software (SPSS Inc., Chicago, IL, USA). Categorical data were presented as numbers of samples and percentages (%), while continuous data were described as mean ± standard deviation (SD). Normal distribution tests were verified using Kolmogorov–Smirnov and Shapiro–Wilk statistics. Independent-samples *t*-test and chi-square test were utilized to compare results for variables normally distributed between groups. A two-tailed *p*-value ˂ 0.05 was defined as statistically significant.

## 3. Results

### 3.1. General Characteristics of the Participants

A total of 280 workers were included in this study. The labor intensity of the control group was similar to that of the exposed group, but individuals in the exposed group were exposed to areas that had statistically higher levels of PM_2.5_ than those in the control group (1.28 ± 0.36 vs. 0.46 ± 0.13 mg/m^3^, *p* < 0.001). No age, working age, BMI, and smoking rate differences existed between these two groups (Table 1).

### 3.2. Lung Function Test Indicators between Exposed Group and Control Group

The pulmonary function test indicators FEV1.0, MMEF, PEFR, PEFR%, FEF75, FEF75%, FEF50%, FEF25, FEF25%, and FEV1.0/FVC% were all lower in the exposed group than in the control group (all *p* < 0.05, Table 2).

### 3.3. Lung Function Damage in the Exposed Group and the Control Group

According to the lung function indicator measurements, lung function damage was divided into restrictive (FVC < 80 and FEV1.0 > 70), obstructive (FVC > 80 and FEV1.0 < 70), and mixed (FVC < 80 and FEV1.0 < 70) respiratory dysfunction, and was further divided into mild, moderate, and severe according to the extent. The results show that the main respiratory dysfunction was restrictive (Figure 1A), and the prevalence rate was higher in the exposed group than in the control group (41.4% vs. 23.6%). Obstructive respiratory dysfunction and mixed respiratory dysfunction were found in only a few of the participants, and no difference was found between the exposed group and control group for either obstructive nor mixed respiratory dysfunction. After further analysis by subdividing the lung damage categories, we found that “no” and “mild” damages were commonly observed in those with restrictive respiratory dysfunction in both the exposed and control groups, and a significant difference was found between the exposure and control groups (χ^2^ = 10.381, *p* = 0.016, Figure 1B).

### 3.4. ECG Abnormalities in the Exposed Group and the Control Group

According to the ECG detection, the main abnormal ECG was sinus bradycardia in the exposed and control groups. There were 14 (10.0%) cases of sinus bradycardia in the exposed group and 9 (6.4%) cases in the control group, but no statistically significant difference was found (χ^2^ = 1.184, *p* = 0.276, Figure 2).

### 3.5. Hypertension in the Exposed Group and the Control Group

According to the diagnostic criteria of hypertension of the World Health Organization (WHO), there were 18 (12.9%) cases of hypertension in the exposed group and 10 (7.1%) cases in the control group, but no statistically significant difference was found (χ^2^ = 2.540, *p* = 0.111, Figure 2).

## 4. Discussion

This study evaluated the lung function, ECG, and blood pressure of dust-contacted workers in the working environment of an open-pit manganese mine by PM_2.5_ particles exposure assessment. Our results suggest that the FVC, FEV1.0, PEFR, and FEF75 of the exposed workers were obviously affected, which indicates that high concentrations of PM_2.5_ particles from the production environment have damaged the lung function of the exposed workers, with restrictive expiratory obstacles as main concern. Long-term exposure to PM_2.5_ has been demonstrated to decrease FEV1.0 and the FVC, accelerating the decline in lung function in healthy adults in a population-based cohort study [14]. Both the large and small airways have been damaged inthe PM-exposed workers, with lung function damage mainly consisted in the main (large) airway. Due to its narrow ducts, large branches, and slow flow velocity, the small airway representsonly 20% of the obstruction in airway resistance. FEF25 and FEF50 are commonly used to determine the incidence of small airway obstruction. In this study, the FEF25 of the exposed workers was also decreased significantly, which suggests that the small airway has also been damaged.

The effects of industrial dust in the working environment on the lung function of the workers are closely related to dust dispersion degree, particle size, and free silica content. Inhalable dust, high total dust concentration, and high free silica contents are more harmful to the human body [15]. When productive dust enters into the human body, around 98% of the dust particles that enter the respiratory tract can be eliminated from the body by self-purification functions, buttheremaining2% of the inhalable dust particles (such as PM_2.5_) will stay in the lung and eventually damage the human body [16]. The inhalable PM_2.5_ may enter the bronchioles and terminal alveoli, and eventually become engulfed by macrophages, staying in bronchial and alveolar cells for a long time, resulting in bronchial and alveolar stenosis, occlusion, and inflammation, and further oppressing bronchial smooth muscle and elastic fibers, thus potentially causing damage to the human body [17,18]. Moreover, inhalable PM_2.5_ may directly affect the respiratory tract by causing irritation and/or allergic reactions in some people, which results in increases in inflammatory secretions caused by reactive airway dysfunction syndrome [19,20]. Furthermore, the allergies can cause bronchial wall spasms, eventually causing airway stenosis [21]. These effects cause increases in alveolar surface tension, with reductions in lung compliance, resulting in lung capacity reduction, which is the main result of lung function damage. It has previously been shown that type II alveolar epithelium was damaged in the process of inflammation stimulation fibrosis, and therefore lung surface active substance synthesis was reduced, or the secretion was insufficient, or the surface-active substance was mostly destroyed and consumed [22].

When pulmonary alveoli are heavily damaged, and capillary wall becomes increasingly thick, the lung can develop diffuse dysfunction as a result. Therefore, the types of lung function damage commonly found in dust-exposed workers are often restrictive ventilatory dysfunction [23], which is consistent with this study. At the same time, the early damage of inhalable PM_2.5_ to the cell membrane can change the permeability of the membrane, which is the main cause of death of alveolar macrophages [24,25]. When macrophages are damaged, the immunity of the respiratory system will be reduced, eventually leading to pulmonary fibrosis and pulmonary heart disease; and the blocking of minute vessels can cause blood pressure to increase simultaneously, which is the cause of cardiovascular disease by exposure to fine particulate matter pollution.

When abnormal small airway function occurs, as a result of the strong compensatory function of the lung, clinical presentations of conventional pulmonary ventilation function can still be normal. Abnormal small airway function of spirometry was reflected in the maximal expiratory flow volume (MEFV) curve, which appeared approximately normal for high volume patterning, but appeared pitted for low volume patterning. The forced vital capacity and time vital capacity were still normal, the maximum expiratory flow rate and FEF25 were basically normal, but FEF50 and FEF75 decreased. The results of lung function detection show that changes to FEF75 and FEF25 were more obvious, while FEF50 showed no obvious change. However, as a result of the short exposure time to PM of the workers, and the function of small airway lesions in early reversibility, the degree to which FEF75 functions as a sensitive index for evaluating small airway damage still warrants more research [26]. Pulmonary ventilation function has played an important role in the evaluation of the effects of air pollution on the respiratory system and can serve as an early indicator in regard to lung diseases.

The PM_2.5_ in the working environment of the open-pit manganese mine has been shown to be harmful to the lung function of the exposed workers, and the harm is predominantly in the form of restrictive expiratory disorder. A large number of studies have shown that the surface of PM_2.5_ adsorbs a large amount of organic substances and inorganic heavy metals [11]. PM_2.5_ pollution contributes to increased mortality from cardiovascular and respiratory diseases [27]. However, the dust exposure duration of the workers in this study has not yet associated with cardiovascular disease, and the early influence of PM_2.5_ exposure on the cardiovascular system still needs to be further studied.

The results also show that the main abnormal ECG was sinus bradycardia for open-pit manganese mine workers. However, there was no statistical difference in the abnormal ECG rate between the exposed group and the control group. A meta-analysis study indicated that ambient PM_2.5_ exposure was associated with a high risk of atrial fibrillation [28], and another study suggested that atrial fibrillation was not significantly associated with short-term exposure to PM_2.5_ air pollution [29]. However, few studies have explicitly studied the relationship between sinus bradycardia and PM_2.5_ so far. Therefore, our study is preliminary, and whether there is no significant difference caused by the short exposure time or a low correlation between sinus bradycardia and PM_2.5_ is still uncertain. Additionally, there was no statistical difference in blood pressure between the exposed group and the control group; the influence of dust exposure on blood pressure was also uncertain in this study. Blood pressure is affected by many factors, such as genetics, obesity, smoking, alcohol consumption, long-term mental stress, and other factors [30]. Studies have shown that work-related stress can cause blood pressure to increase [31,32]. In the mine that we considered forour study, which applies a four-shift/three-shift system of shift work, day and night shifts may also result in the mine workers experiencing increases in blood pressure.

It should be noted that this was a cross-sectional study, and the duration of the PM_2.5_ exposure of the mining workers was not very long; as such, we cannot directly identify a clear dose–effect relationship between cardio-pulmonary function and PM_2.5_ exposure, which is a limitation of the study. In addition, the controls also came from the mining workers in the same mine/factory. Although their exposure levels were relatively low, a certain PM_2.5_ exposure may have reduced the study’s statistical efficacy. Other workers from areas with cleaner air may serve as more appropriate controls. Therefore, the PM_2.5_-exposed workers need to be tracked in future investigations, with periodic PM_2.5_ concentration measures and testing for lung function, electrocardiogram, blood pressure, and other indicators testing, so as to better allow for the prevention and early treatment of potential occupational diseases.

## 5. Conclusions

We investigated the effects of PM_2.5_ exposure on the cardio-pulmonary function of 280 dust-exposed workers in an open-pit manganese mine. The PM_2.5_ exposed group had significantly lower values of pulmonary function indexes of FEV1.0, MMEF, PEFR, PEFR%, FEF75, FEF75%, FEF50%, FEF25, FEF25%, and FEV1.0/FVC%. Our results suggested that PM_2.5_ exposure resulted in harm to the pulmonary function of workers in the open-pit manganese mine, and the most common injury was restrictive ventilatory disorder. Electrocardiogram (ECG) abnormalities, especially sinus bradycardia, appeared in both groups, but the prevalence rate was not significantly different between the groups. The early effect of PM_2.5_ exposure on the cardiovascular system was uncertain at the exposure levels and exposure time documented in this study, therefore further long-term follow-up studies are required.

## Figures and Tables

**Figure 1 ijerph-16-02017-f001:**
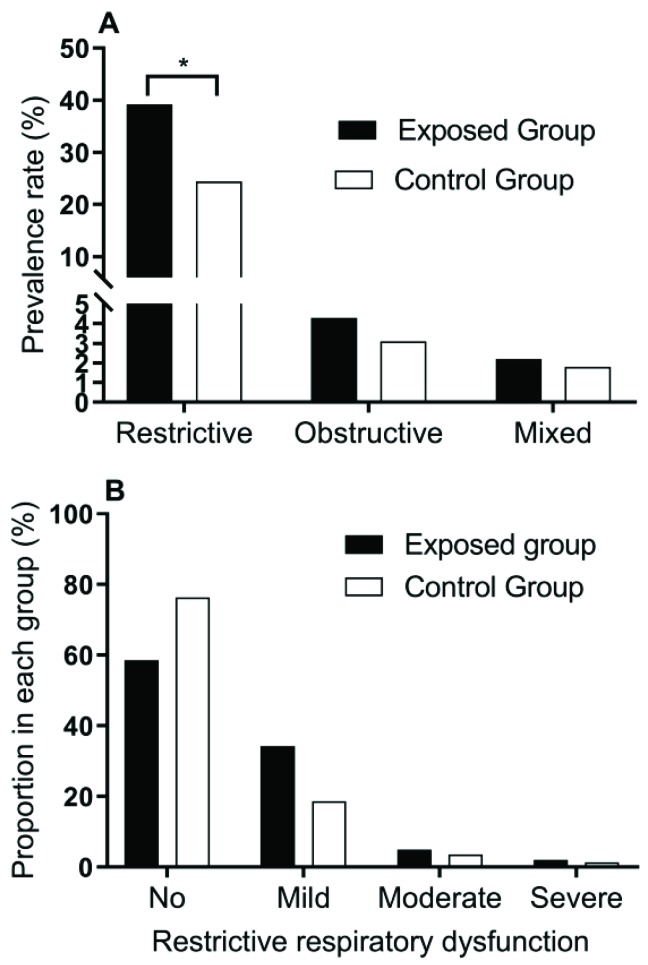
Distribution of different types of respiratory dysfunction (**A**) and proportions of different restrictive respiratory dysfunction levels (**B**) between the exposed group and control group. * *p* < 0.05.

**Figure 2 ijerph-16-02017-f002:**
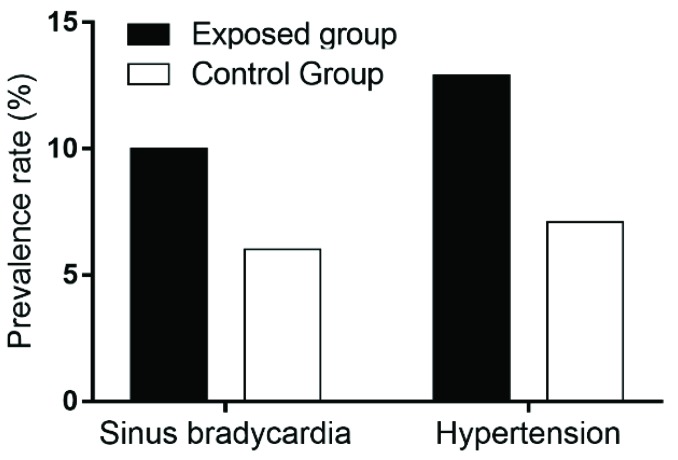
Prevalence rate of sinus bradycardia and hypertension between the PM_2.5_ exposed group and control group.

**Table 1 ijerph-16-02017-t001:** General characteristics of the participants.

Characteristics	Exposed Group (*n* = 140)	Control Group (*n* = 140)	*t or χ* ^2^	*p*
Age (years)	32.40 ± 7.53	31.21 ± 6.83	1.385	0.167
Working age (years)	2.28 ± 0.50	2.18 ± 0.96	1.093	0.275
Sex (*n*, %)				
Male	121 (86.4)	110 (78.6)	2.993	0.084
Female	19 (13.6)	30 (21.4)	1.088	0.278
Weight (kg)	66.65 ± 6.42	65.83 ± 6.19	1.723	0.086
Height (cm)	168.35 ± 7.29	166.74 ± 8.31	0.988	0.324
BMI (kg/m^2^)	24.04 ± 3.39	23.63 ± 3.55	3.527	0.060
Smoking (*n*, %)	68 (48.6)	52 (37.1)	25.35	<0.001
PM_2.5_ concentration of 8-h TWA in the working places (mg/m^3^)	1.28 ± 0.36	0.46 ± 0.13		

BMI: body mass index; 8-h TWA: eight-hour time-weighted average concentration.

**Table 2 ijerph-16-02017-t002:** Lung function indicators between the PM_2.5_ exposed group and control group.

Indexes	Exposed Group (*n* = 140)	Control Group (*n* = 140)	*t*	*p*
FVC (liter)	3.41 ± 0.99	3.65 ± 1.22	−1.807	0.072
FVC%	90.17 ± 16.70	91.98 ± 24.06	−0.731	0.465
FEV1.0 (liter)	3.28 ± 0.95	3.54 ± 1.13	−2.084	0.038
FEV1.0%	100.21 ± 17.89	103.61 ± 26.38	−1.262	0.208
MMEF (liter/second)	5.69 ± 2.14	6.28 ± 1.90	−2.439	0.035
PEFR (liter/second)	8.97 ± 3.27	9.93 ± 3.02	−2.552	0.011
PEFR%	104.04 ± 27.80	110.89 ± 26.73	−2.102	0.036
FEF75 (liter/second)	4.21 ± 1.78	4.63 ± 1.67	−2.036	0.043
FEF75%	206.14 ± 81.03	235.57 ± 85.31	−2.960	0.003
FEF50 (liter/second)	6.90 ± 2.63	7.45 ± 2.32	−1.856	0.065
FEF50%	151.81 ± 49.70	164.54 ± 47.60	−2.189	0.029
FEF25 (liter/second)	8.76 ± 3.28	9.54 ± 3.03	−2.067	0.040
FEF25%	122.82 ± 36.47	131.43 ± 34.31	−2.035	0.043
FEF50/FEF25	80.44 ± 15.82	79.50 ± 14.17	0.524	0.601
FEV1.0/FVC%	111.64 ± 0.09	113.10 ± 0.07	−151.511	<0.001
MVV	135.42 ± 35.57	144.02 ± 45.45	−1.763	0.079

FVC: forced vital capacity; FEV1.0: forced expiratory volume in one second; MMEF: maximum mid expiratory flow; PEFR: peak expiratory flow rate; FEF: forced expiratory flow; MVV: maximum ventilatory volume.

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
