# Peer review of "Effects of PM2.5 on Cardio-Pulmonary Function Injury in Open Manganese Mine Workers"

_ijerph, 2019, doi:10.3390/ijerph16112017_

Reviewer 1 Report

This manuscript explored PM2.5 exposure level and cardio-pulmonary function among mine workers. Classifying exposed groups and unexposed groups, the authors found that PM2.5 exposure level was higher among exposed group compared to unexposed group. Furthermore, lung function was weak as well as prevalence of restrictive respiratory dysfunction was high among exposed group. They did not find any significant difference for cardiovascular system.

Overall, the manuscript is easy to follow. Research question is clear, method is straightforward, and results are presented well. Although the results are somewhat expected to see, this will add the current understanding regarding the health risk of mine workers. I have several suggestions to improve the manuscript.

[Abstract] “between two groups (41.4% vs 23.6%, p=0.016)” In the result section, you mentioned this is 39.3% vs 24.5%.

[Abstract] “PM2.5 exposure did harmful to pulmonary function of workers.” It is recommended to avoid causal term in this kind of study setting. While you found some relationships, this does not support causation between PM2.5 exposure and pulmonary function. Rephrase this sentence with taking out harmful.

[Introduction] “Studies have shown higher mortality rates from PM2.5 exposure due to a variety of disease, including lung cancer (23.9%).” This interpretation of Song et al. is wrong and needs to rephrase.

[Introduction] Some more previous works focusing on PM2.5 exposure among miners are necessary. The authors put several citations for PM2.5 and health among general population, but these relationships are not same among miners, as the authors also pointed out.

[Introduction & Discussion] Put several unique points of this study to highlight the difference from the previous studies.

[Methods] Some more background info is necessary about mine. Where (or which province) is this located? How large this mine is?

[Methods] More text is necessary regarding time-weighted average. How did you weight by time?

[Methods] More text is necessary regarding health examination. When is this exam taken place? Is this right after workers working hour, or middle/late of working hour? This might be a potential source of bias.

[Results] “PM2.5 particles in the environmental have certain damage to lung function.” This sentence is implying causation, and this kind of words should be avoided (see above). Also, the results section should focus on the result, and this kind of statement should be mentioned in the discussion section.

[Figure 1] I assume ‘*’ implies statistically significant level, but there is no explanation about this indication. In Figure 1B, it is not clear which comparison is P= 0.016, and needs revision.  

[Editorial]All “2.5” of PM2.5 should be expressed in subscript.  

[Editorial] Some sentences need English grammar editing.

Author Response

Reviewer #1

Comments and Suggestions for Authors

This manuscript explored PM2.5 exposure level and cardio-pulmonary function among mine workers. Classifying exposed groups and unexposed groups, the authors found that PM2.5 exposure level was higher among exposed group compared to unexposed group. Furthermore, lung function was weak as well as prevalence of restrictive respiratory dysfunction was high among exposed group. They did not find any significant difference for cardiovascular system.

Overall, the manuscript is easy to follow. Research question is clear, method is straightforward, and results are presented well. Although the results are somewhat expected to see, this will add the current understanding regarding the health risk of mine workers. I have several suggestions to improve the manuscript.

[Abstract] “between two groups (41.4% vs 23.6%, p=0.016)” In the result section, you mentioned this is 39.3% vs 24.5%.

Response: Sorry for the confusing and mistake. After carefully checking, the result should be 41.4% vs 23.6%, p=0.016. Therefore, the information in the result section was corrected.

[Abstract] “PM2.5 exposure did harmful to pulmonary function of workers.” It is recommended to avoid causal term in this kind of study setting. While you found some relationships, this does not support causation between PM2.5 exposure and pulmonary function. Rephrase this sentence with taking out harmful.

Response: Thanks for the suggestions. The sentence was changed to: “PM2.5 exposure was associated with pulmonary function damage of workers in the open-pit manganese mine…”. Line 24-25

[Introduction] “Studies have shown higher mortality rates from PM2.5 exposure due to a variety of disease, including lung cancer (23.9%).” This interpretation of Song et al. is wrong and needs to rephrase.

Response: The sentence was changed to: “One study conducted in China suggested that PM2.5 in 2015 contributed as much as 40.3% to total stroke deaths, 33.1% to acute lower respiratory infection (<5yr) deaths, 26.8% to ischemic heart disease (IHD) deaths, 23.9% to lung cancer (LC) deaths, 18.7% to chronic obstructive pulmonary disease (COPD) deaths, 30.2% to total deaths combining IHD, stroke, COPD, and LC, and 15.5% to all cause deaths.” Line 65-69

[Introduction] Some more previous works focusing on PM2.5 exposure among miners are necessary. The authors put several citations for PM2.5 and health among general population, but these relationships are not same among miners, as the authors also pointed out.

Response: Thanks for suggestion! Compared with PM2.5 and health among general population, few studies were conducted among miners. After literature retrieval again, we added the following citation: [Leon-Kabamba N, Ngatu NR, Kakoma SJ, Nyembo C, Mbelambela EP, Moribe RJ, Wembonyama S, Danuser B, Oscar-Luboya N. Respiratory health of dust-exposed Congolese coltan miners. Int Arch Occup Environ Health. 2018; 91(7): 859-864.]. And the following information was added to replace the redundancy of the origin: Another study showed high prevalence of respiratory complaints in Congolese informal coltan miners, and an inverse association was observed between lung function (peak expiratory flow rate, PEFR) and PM2.5 exposure. Line 73-75

[Introduction & Discussion] Put several unique points of this study to highlight the difference from the previous studies.

Response:

We investigated the effects of PM2.5 exposure on the cardio-pulmonary function of 280 dust-exposed workers in an open-pit manganese mine, which is different from the general population.

Effects of PM2.5 exposure on the cardio-pulmonary function of mine workers are also scarcely investigated.

The chemical compositions of PM2.5 from air pollution and mining may be also different, the health effects of PM2.5 from air pollution have been well studied, but few studies focused on PM2.5 from mining.

[Methods] Some more background info is necessary about mine. Where (or which province) is this located? How large this mine is?

Response: The following sentences were added to introduce the background information: The manganese mine is located in Raoping county, Chaozhou city, Guangdong province of China, with a latitude of 23.70 O N and longitude of 116.93 O E. The manganese mine is moderate-sized in China with a total area of 150 acres, included several workshops….. Line 97-101

[Methods] More text is necessary regarding time-weighted average. How did you weight by time?

Response: time-weighted average of PM2.5 was calculated by the following formula:

TWA=E/Ttotal, E=C1T1+C2T2+……CnTn, where Ttotalmeans the working time in a working day, and is 8h in this study; E is the PM2.5 exposure dose under working hours; Cn is the corresponding concentration of contact in the time period of Tn; and Tn is the corresponding contact duration under the concentration of Cn. The PM2.5concentration was determined for successive 7 days, therefore n is 7 in this study.

   The information was also added to the text. (Line 103-127)

[Methods] More text is necessary regarding health examination. When is this exam taken place? Is this right after workers working hour, or middle/late of working hour? This might be a potential source of bias.

Response: Thanks! The health examinationfor all the participants was performed in the morning before working. This sentence was added to the text. Line 152-153

[Results] “PM2.5 particles in the environmental have certain damage to lung function.” This sentence is implying causation, and this kind of words should be avoided (see above). Also, the results section should focus on the result, and this kind of statement should be mentioned in the discussion section.

Response: This sentence has been deleted from the Results section.

[Figure 1] I assume ‘*’ implies statistically significant level, but there is no explanation about this indication. In Figure 1B, it is not clear which comparison is P= 0.016, and needs revision. 

Response: Thanks for reminding! Yes, *’ implies statistically significant level, and the explanation was added to the figure legend. In Figure 1B, P=0.016 refer to the significant difference in the proportion of the different levels of restrictive respiratory dysfunction between the exposed group and control group. This information is now described in the text, therefore “P=0.016” was deleted from Figure 1B.

[Editorial]All “2.5” of PM2.5 should be expressed in subscript.

Response: All “2.5” of PM2.5 has been expressed in subscript throughout the manuscript.

[Editorial] Some sentences need English grammar editing.

Response: The manuscript has been checked carefully by colleagues for English grammar editing, and many revisions have been made.

Reviewer 2 Report

The current study is mostly descriptive. The data on lung function is interesting. However, the results are not well described. Below are the specific comments:

Both Figures 1 and 2 lack error bars. 

Even though authors describe Restrictive and Obstructive respiratory dysfunction in methods, it will be helpful for readers to have that information in results.

Author Response

Reviewer #2

Comments and Suggestions for Authors

The current study is mostly descriptive. The data on lung function is interesting. However, the results are not well described. Below are the specific comments:

Both Figures 1 and 2 lack error bars. 

Response:Figure 1 is the distribution of different types of respiratory dysfunction and proportions of different restrictive respiratory dysfunction levels between the exposure group and control group. It stands for prevalence or proportion of the data, which is only a value for each type, so there are no error bars.

Figure 2 stands for the prevalence rate of sinus bradycardia and hypertension between the exposure group and control group, which is also a value for each type, so there are no error bars.

Even though authors describe Restrictive and Obstructive respiratory dysfunction in methods, it will be helpful for readers to have that information in results.

Response:The details for defining Restrictive and Obstructive respiratory dysfunction were added to the results section: “……lung function damage was divided into restrictive (FVC<80 fev1.0="">70), obstructive (FVC>80 & FEV1.0<70) and mixed (FVC<80 & FEV1.0 <70) respiratory dysfunction……” Line 230-233

Reviewer 3 Report

Congratulations for an interesting job. The authors linked data on exposure to PM2,5 with data identifying the health of employees (cardiovascular and respiratory capacity). I have a question for the authors, please explain when spirometric measurements were made (eg as part of the periodic examination of employees) and what instrument? Were they made at the same time (season), were they made by one, the same spirometrist or several? Similarly for pressure measurements and ECG tests (as an ECG instrument). Was the medical history of the current state of health collected during the tests, if so what were the results? The discussion lacked critical discussion (even 1-2 sentences) about the potential impact of a slightly higher frequency of smokers (though statistically insignificant p = 0.06) in the study group compared to controls.

Author Response

Reviewer #3 Comments and Suggestions for Authors

Congratulations for an interesting job. The authors linked data on exposure to PM2,5 with data identifying the health of employees (cardiovascular and respiratory capacity). I have a question for the authors, please explain when spirometric measurements were made (eg as part of the periodic examination of employees) and what instrument? Were they made at the same time (season), were they made by one, the same spirometrist or several? Similarly for pressure measurements and ECG tests (as an ECG instrument). Was the medical history of the current state of health collected during the tests, if so what were the results? The discussion lacked critical discussion (even 1-2 sentences) about the potential impact of a slightly higher frequency of smokers (though statistically insignificant p = 0.06) in the study group compared to controls.

Response: The health examination for all the participants was performed in the morning before working. All the examination was performed at the same time by several professional medical workers according uniform standard. Spirometric measurements were conducted in the morning before working by professional medical workers using portable pulmonary function test apparatus (AS-507, Minato, Japan). The routine 12-leads electrocardiogram (ECG) was performed by professional medical staff using an electrocardiograph (PC-80A, Heal Force, China). Please see also line 152-153, line 161-162, line 167-168

Reviewer 4 Report

This is an interesting study, I have a few comments for the authors to consider:

PM2.5: fine particulate matter pollution

PM2.5 -- PM2.5

In Title, please use PM2.5

I would like the authors to clarify the time of exposure assessment and the health status.

Even based on a cross-sectional study as discussed by the authors, it is still fine to examine the dose-response relationship.

 what is the criteria to divid the two group: exposure and control groups.

The details of PM2.5 assessment are needed

I don't think it is a simple random selection, please revise.

One major issue: the authors need to conduct regression analysis to control for other covariates.

Author Response

Reviewer #4

Comments and Suggestions for Authors

This is an interesting study, I have a few comments for the authors to consider:

PM2.5: fine particulate matter pollution

Response:The phrase “particulate matter” was changed to: “fine particulate matter pollution”.

PM2.5 -- PM2.5

Response:Throughout the manuscript, the phrase “PM2.5” was changed to “PM2.5”.

In Title, please use PM2.5

Response: Thanks for the suggestion! The title was changed to: “Effects of PM2.5 on cardio-pulmonary function injury in open manganese mine workers”.

I would like the authors to clarify the time of exposure assessment and the health status.

Response:As response to Reviewer #1, the health examinationfor all the participants was performed in the morning before working. This sentence was added to the text.

For the exposure assessment, we used the 8 hours time-weighted average (TWA) of PM2.5, which was calculated by the following formula:

TWA=E/Ttotal, E=C1T1+C2T2+……CnTn, where Ttotalmeans the working time in a working day, and is 8h in this study; E is the PM2.5 exposure dose under working hours; Cn is the corresponding concentration of contact in the time period of Tn; and Tn is the corresponding contact duration under the concentration of Cn. The PM2.5concentration was determined for successive 7 days, therefore n is 7 in this study.

All these information was added to the revised manuscript. Please see also line 103-127

Even based on a cross-sectional study as discussed by the authors, it is still fine to examine the dose-response relationship.

Response: Thanks for the suggestions! Just as you said, it is fine to examine the dose-response relationship between PM2.5 exposure and health status. But in this study, we only measured the PM2.5 concentration in the different workshops, but not individual exposure level. Therefore, we cannot evaluate the dose-response relationship based on individual exposure levels. In our future work, we will measure the PM2.5 exposure level for individual worker and therefore we can evaluate the dose-response relationships.

what is the criteria to divide the two group: exposure and control groups.

Response: As described in the manuscript, the PM2.5 concentration was measured by TSI SIDEPAK AM510 Aerosol Monitor in different workshops, the belt driving, mineral crushing, and mineral sifting workshops in the manganese mine have higher PM2.5 concentration than the mineral separation workshop, therefore,the workers were divided into the exposure group and the control group according to the workshops where they served.

The details of PM2.5 assessmentare needed

Response:Additional information of PM2.5 assessment were provided, and the following paragraph was added: “The PM2.5 concentration of 8h-TWA (Time-weighted average concentration) in these different workshops had been determined by TSI SIDEPAK AM510 Aerosol Monitor (TSI Inc., USA), according to the detailed instruction of the instrument for successive 7 days. The 8h-TWAconcentration of PM2.5 was calculated by the following formula:

TWA=E/Ttotal, E=C1T1+C2T2+……CnTn,

where Ttotal means the working time in a working day, and is 8h in this study; E is the PM2.5 exposure dose under working hours; Cn is the corresponding concentration of contact in the time period of Tn; and Tn is the corresponding contact duration under the concentration of Cn. The PM2.5 concentration was determined for successive 7 days, therefore n is 7 in this study.”.

I don't think it is a simple random selection, please revise.

Response:Sorry for the confusing! This study is not a simple random selection. And we have revised the method description to clarify. Please see line 128-135.

One major issue: the authors need to conduct regression analysis to control for other covariates.

Response: Thanks for the suggestion! But in this study, we only measured the PM2.5 concentration in the different workshops, but not individual exposure level. Therefore, we cannot conduct regression analysis to control for other covariates based on individual exposure levels, but compare the differences between the two groups. In our future work, we will measure the PM2.5 exposure level for each individual worker and therefore we can do this work.

Round  2

Reviewer 4 Report

The authors have done a goog job in revising the manuscript, I would suggest to accept it.